# Foregut organ progenitors and their niche display distinct viscoelastic properties in vivo during early morphogenesis stages

Aliaksandr Dzementsei [1,3], Younes F. Barooji [2,3], Elke A. Ober [1✉] & Lene B. Oddershede [2✉]

Material properties of living matter play an important role for biological function and development. Yet, quantification of material properties of internal organs in vivo, without causing physiological damage, remains challenging. Here, we present a non-invasive approach based on modified optical tweezers for quantifying sub-cellular material properties deep inside living zebrafish embryos. Material properties of cells within the foregut region are quantified as deep as 150 μm into the biological tissue through measurements of the positions of an inert tracer. This yields an exponent, $\alpha$, which characterizes the scaling behavior of the positional power spectra and the complex shear moduli. The measurements demonstrate differential mechanical properties: at the time when the developing organs undergo substantial displacements during morphogenesis, gut progenitors are more elastic ($\alpha = 0.57 \pm 0.07$) than the neighboring yolk ($\alpha = 0.73 \pm 0.08$), liver ($\alpha = 0.66 \pm 0.06$) and two mesodermal ($\alpha = 0.68 \pm 0.06$, $\alpha = 0.64 \pm 0.06$) progenitor cell populations. The higher elasticity of gut progenitors correlates with an increased cellular concentration of microtubules. The results infer a role of material properties during morphogenesis and the approach paves the way for quantitative material investigations in vivo of embryos, explants, or organoids.

[1] Novo Nordisk Foundation Center for Stem Cell Biology, University of Copenhagen, Blegdamsvej 3b, 2200 Copenhagen N, Denmark. [2] Niels Bohr Institute, University of Copenhagen, Blegdamsvej 17, 2100 Copenhagen, Denmark. [3]These authors contributed equally: Aliaksandr Dzementsei, Younes F. Barooji. ✉email: elke.ober@sund.ku.dk; oddershede@nbi.ku.dk

Material properties are crucial for action–reaction mechanisms and are closely linked to motion, also within living organisms. Viscoelasticity, short for the combined viscous and elastic properties of a material, determines the fluidity and stiffness of a material and the way it responds to internally and externally generated forces[1]. Furthermore, viscoelasticity has been shown to correlate, e.g., with the invasiveness of cancer[2] and with the differentiation of stem cells[3,4]. The ability of cells to move, maintain, or change their shape is in part regulated by their reaction to external forces depending on the cellular material properties[5]. However, the in vivo material properties of cells, tissues or organs are largely unknown and quantification of these is the first step towards understanding how material properties may contribute to organ and overall embryo morphogenesis.

The contribution of mechanical forces and biophysical material properties, such as viscoelasticity, remains poorly understood, predominantly due to a lack of tools to measure and quantify material properties within complex biological systems in vivo. Several assays have been developed to analyze microrheological properties of cells cultured in vitro or of surface embryonic tissues, such as atomic force microscopy, micro-aspiration or optical tweezers[6]. These assays can be used to quantify the viscous and elastic properties of living matter on a variety of time and length scales down to micro-seconds and the subcellular level[7–9]. However, in vivo investigation of material properties of cells and tissues forming internal organs remains difficult due to the challenge of performing accurate quantitative measurements, especially deep within a living organism without causing physiological damage to the investigated organism.

Optical tweezers are widely used in vitro to investigate protein folding[10], and also to quantify material properties of isolated cells[11]. In its simplest form, an optical trap is formed by tightly focusing a laser beam, whereby objects with a refractive index higher than the surroundings are drawn along the intensity gradient towards the focus of the beam by a harmonic force[12]. In water, the use of adaptive optics has allowed for the optical trapping of particles as deep as 166 μm[13].

The near-infrared (NIR) biological transparency window allows NIR lasers to penetrate deeply into biological tissues and if NIR-based optical traps are operated at sufficiently low laser powers, they cause no observable physiological damage[14]. There have been reports of successful optical manipulation of particles inside zebrafish[15,16], where also red blood cells[17] and otoliths have been reported trapped, the latter 150 μm inside living zebrafish embryos[18]. However, most of these studies were qualitative and did not provide quantitative information about the material properties at the location of the trapped particle. In the early Drosophila embryo, optical tweezers were used to probe the mechanics of cell contacts close to the surface of the organism by observing the equilibration of the interface following a deformation[19]. A quantitative approach combining active oscillations and passive measurements was developed based on optical tweezers to determine viscous and elastic parameters inside living cultured cells[20]. A similar method was used to determine the ratio between the viscous and elastic moduli within cells close to the surface of early zebrafish embryos at 5–7 h post fertilization (hpf)[16]. Other approaches to infer viscoelasticity inside living organisms are based on video recordings of either magnetically responsive ferrofluid microdroplets in zebrafish tailbuds[21] or thermal fluctuations of particles in the syncytium of the early Drosophila embryos[22]. These methods were used to determine viscoelastic properties relatively close to the surface of the embryo and cannot directly be applied to investigate internal tissues and organs in vivo due to light scattering and poor resolution deep inside biological tissues.

Here, we measure material properties at subcellular resolution in deep tissues using laser tracking and photodiode detection of the thermal fluctuations of optically trapped nanoparticles. To investigate whether material properties differ between cell types undergoing complex morphogenetic movements, we focused on liver and foregut morphogenesis in zebrafish at the time when organ asymmetry is established by rearrangement of four cell populations next to the yolk[23]. The liver and gut progenitor populations arise from the foregut endoderm. After their specification, liver progenitors form an asymmetric organ primordium, the liver bud, by directional cell migration to the left of the midline (Fig. 1a)[24]. In a parallel process, the entire foregut undergoes leftward looping, which is suggested to result from a passive displacement caused by asymmetric movements of the neighboring left and right lateral plate mesoderm (LPM)[25]. These morphogenetic movements ensure the efficient arrangement of the organs within the body, which is crucial for proper physiological function. The mechanical properties of these developing organs are unknown, probably because they are located 120–150 μm deep inside the embryo (Fig. 1b, c)[23,26]. We demonstrate consistent and significant differences in cellular viscoelasticity between the analyzed progenitor populations, with the gut progenitors being the most elastic in comparison to neighboring rearranging tissues. The timescales probed in the current investigation are those relevant for cytoskeletal biopolymer dynamics inside living cells and our data show that the observed biomechanical differences correlate with different concentrations of microtubules. Hence, material properties may influence morphogenetic tissue movements during organ formation and embryonic development.

## Results

**Microinjected nanoparticles can be optically trapped deep within living zebrafish embryos.** Viscoelastic properties of a matrix, such as the intracellular environment, can be mapped by optical tweezers-based tracking of thermal fluctuations of particles within the matrix[7,8,27]. Intracellular organelles like lipid granules can serve as trackable endogenous particles and have successfully been used for probing single-cell systems[8,28]. However, in vivo tracking of lipid granules is difficult in the majority of deep tissues as their signal-to-noise ratio is low and decreases further as penetration depth increases due to increasing spherical aberration.

In our study, we assessed the suitability of 0.2 μm gold and 0.5, 0.8, and 1 μm fluorescently labeled polystyrene nanoparticles (size indicates particle diameters) as tracers and introduced them by microinjection into 1-cell stage zebrafish embryos. During embryonic development, the 0.5-μm polystyrene beads distributed throughout the embryo without causing any apparent morphological defects at 30 hpf (Fig. 1d, e) or 5 days post fertilization, as determined by bright-field stereomicroscopy. In the foregut region, most of these nanoparticles were located intracellularly with typically only one bead per cell positioned between the nucleus and the plasma membrane (Fig. 1f). In contrast, gold nanoparticles, as well as 0.8- and 1-μm polystyrene beads were rarely or not detected in the foregut region. The latter could be due to the large particle size compared to the relatively small progenitor cells in the foregut region.

The optical tweezer experiments were conducted on living zebrafish embryos immobilized in agarose and mounted dorsolaterally on a microscope slide with the liver bud facing the microscope objective (Fig. 2a). In this configuration, embryos develop normally without any visible defects, and could be recovered after 3–4 h. The optical trap was formed by a tightly focused NIR laser (1064 nm) implemented in a confocal microscope (Fig. 2a)[29], thus allowing for both optical trapping and confocal visualization of the fluorescent nanoparticles. The confocal visualization was used to navigate inside the embryos

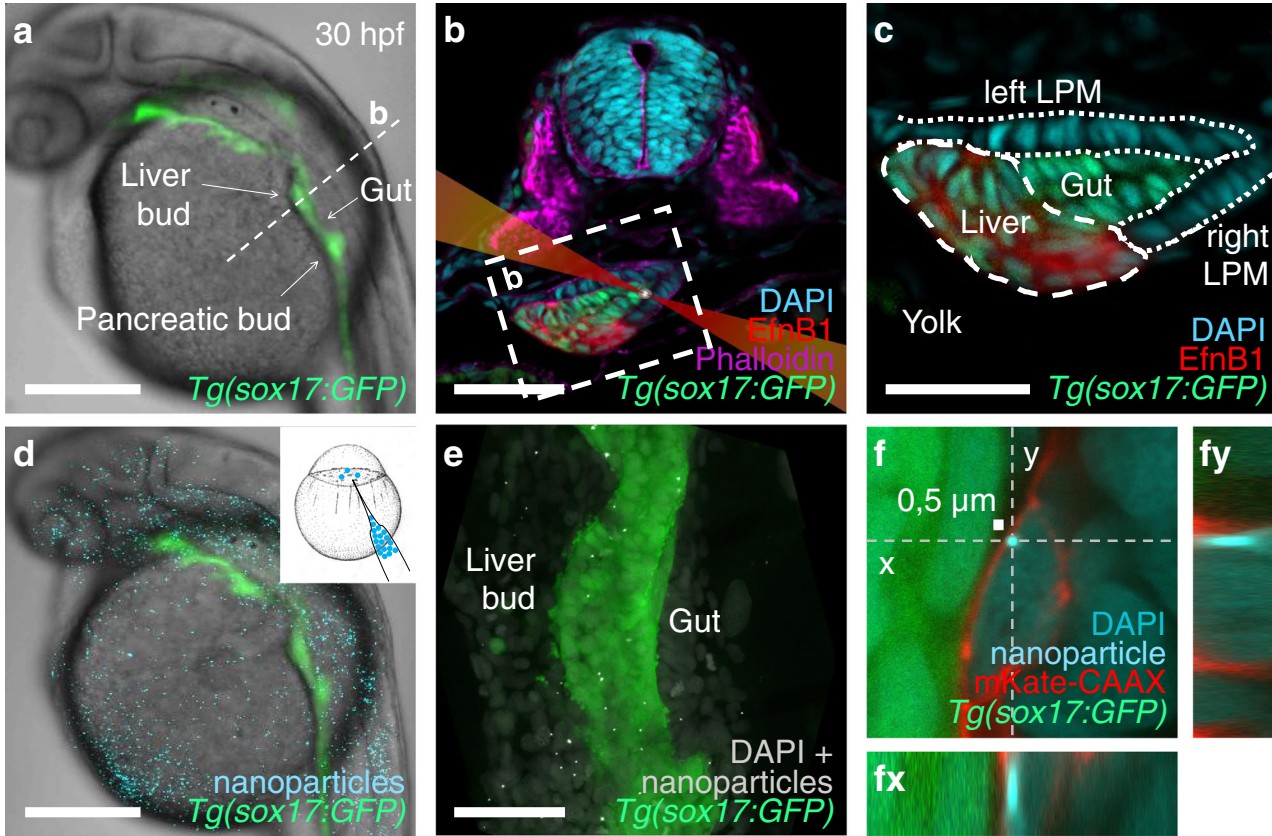

**Fig. 1 Injected nanoparticles distribute throughout the embryo, including the foregut region in 28–30 hpf zebrafish embryos. a** Transgenic *tg(-0.5 sox17:GFP)^zf99* visualizes the endoderm in a living zebrafish embryo, including the foregut organ primordia. **b** Transverse section through the foregut region (dashed line in a) with schematized laser beam path (red) and bead (white). The foregut endoderm expresses transgenic *sox17:GFP* (green) and liver progenitors EphrinB1 (red); DAPI marks nuclei (cyan); and Phalloidin marks F-actin (magenta). **c** Magnification of the foregut region shown in (**b**) (white dashed rectangle). **d** 0.5 μm polystyrene particles (cyan) microinjected at the one-cell stage distribute throughout the embryo, including the endoderm (green), without causing apparent morphological defects. **e** Maximum intensity projection of confocal z-stack showing nanoparticle distribution (white) in the foregut region, GFP marks the endoderm. **f** Confocal image of a representative nanoparticle (light blue) located in the cytoplasm between the nucleus (darker blue) and plasma membrane (red) in a fixed embryo. (fy) and (fx) show orthogonal views in the "yz" plane (dashed line "y" in (**f**)) and "xz" plane (dashed line "x" in (**f**)), respectively. Scale bars **a**, **d**: 250 μm; **b**, **e**: 50 μm; **c**: 30 μm; **f**: 0,5 μm (scale bar next to bead).

using low laser power and at low resolution, while the confocal lasers were turned off during optical trapping. After the trapping experiments, the embryos were fixed, the yolk was removed, and the subcellular location of the trapped nanoparticle in the foregut region was examined by a different confocal setup providing higher resolution (~250 nm).

The injected nanoparticles could be optically trapped in diverse tissues, including somites (muscle cells), spinal cord (neurons), and the foregut region (endoderm and lateral plate mesoderm). To demonstrate optical trapping of nanoparticles in the foregut region, the embedded zebrafish embryo was moved at low speed by a piezo stage relative to the trapping laser. When an internalized particle co-localized with the optical trap, it became visibly trapped and remained trapped for several seconds in spite of the continued movement of the embryo (Supplementary Movie). Likewise, the difference in trajectories of nanoparticles in the zebrafish foregut in trapped or freely diffusing states indicates that the NIR optical tweezers can trap particles at considerable depth in living embryos (inset of Fig. 2b).

### Thermal fluctuations of optically trapped nanoparticles enable quantification of cellular viscoelasticity in internal tissues.

Measurements of the viscoelastic properties of the foregut region were based on monitoring thermal fluctuations of microscopic tracers. This method has been successfully applied to studying the motion of tracers inside single living cells[8,30] as well as for quantifying material properties of in vitro materials[27,31]. To apply this method deep inside the living zebrafish embryo, we co-localized the tightly focused laser, the optical trap, and a nanoparticle in the desired tissue in three dimensions and recorded the positions of the particle's thermal fluctuations by a quadrant photodiode (QPD) without moving the chamber (Fig. 2a), providing a positional resolution of <1 nm. The nanoparticle has a focusing effect on the laser light, therefore the signal detected by the QPD substantially increases when a nanoparticle was in the center of the trap compared to an empty trap within the embryo (Supplementary Fig. 1 and Supplementary Movie). To visually localize the foregut region in the embryo, we utilized transgenic *sox17:GFP* embryos with GFP expression highlighting the endoderm, including the liver and gut progenitors (Fig. 1a). In vivo microrheological measurements were performed using several nanoparticles per embryo. Most measurements were taken at the time of asymmetric liver bud formation, around 30 hpf (Fig. 1a–c), while a subset were acquired between 21 and 23 hpf, just prior to the overt separation into liver and gut progenitors.

After the optical trapping experiments, embryos were fixed and their foregut region was imaged with improved resolution (~250 nm) by a different confocal microscope to determine the location of each of the tracked particles (Fig. 1f). The vast

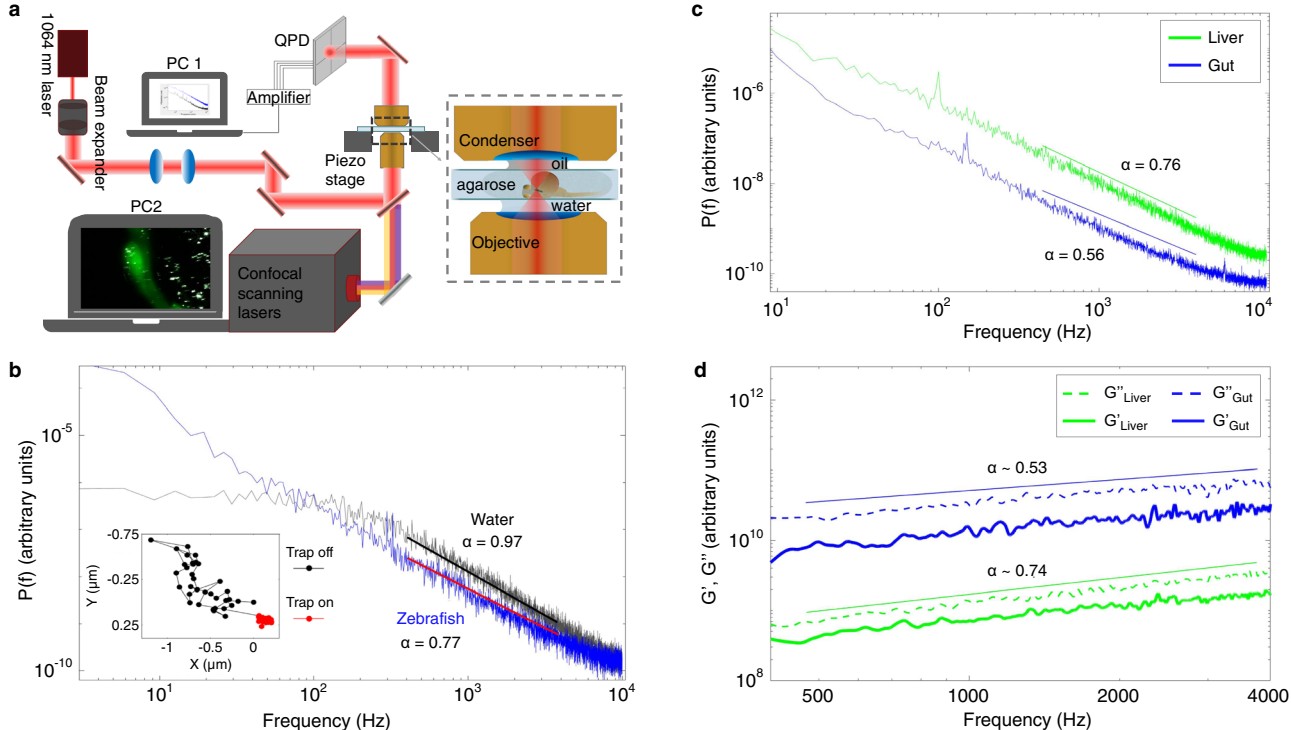

**Fig. 2 Forward-scattered light from optically trapped nanoparticles in living zebrafish can be used to infer cellular viscoelasticity. a** Schematic of the setup for laser tracking of nanoparticles in vivo: the laser beam (1064 nm, red line) is focused in the embryo (inset) and the forward-scattered light is collected by a condenser and imaged onto a quadrant photodiode (QPD). In all optical trapping experiments, the immobilized zebrafish were mounted dorsolaterally in agarose on a microscope slide with the liver bud facing the objective. **b** Examples of positional power spectra of optically trapped particles in water (black, $\alpha = 0.97$) or in a zebrafish embryo (blue, $\alpha = 0.77$). The straight lines show fits of Eq. (1) to data. The inset shows the trajectories of a trapped (red) and a freely diffusing (black) particle in the foregut region of a living zebrafish embryo. **c** Power spectra of optically trapped nanoparticles in liver (green) or gut progenitors (blue). Full lines show linear fits to the double logarithmic plot in the frequency interval 400 Hz < f < 4000 Hz, yielding scaling exponents of 0.56 ± 0.08 (gut) and 0.76 ± 0.06 (liver), respectively. **d** Symbols (blue and green) provide the storage moduli, G', of the gut and liver, same colors as in (**c**) and (**d**). The corresponding loss moduli, G", are shown with dashed lines for comparison of the amplitude. G" dominates G' over the 400–4000 Hz frequency interval for both tissue types. The full lines show fits to the loss moduli data in the corresponding region to the power spectra (**c**), returning scaling exponents of 0.53 ± 04 (gut) and 0.74 ± 0.05 (liver). The data shown in (**c**, **d**) is an average of five experiments for each cell type.

majority of particles was found within the cytoplasm (Supplementary Fig. 2), while particles observed in the extracellular space, as well as particles that could not be clearly assigned were excluded from the analysis.

The power spectrum, $P_x(f)$, of a tracer particle in a matrix can be calculated from its positional time series (Fig. 2c). The power spectrum of a particle in a purely viscous medium, such as water, is well fitted by a Lorentzian function[32,33], consistent with the behavior shown in Fig. 2b with a flat region at low frequencies, reflecting motion where the tracer particle feels the restoring potential of the optical trap, and a scaling region at higher frequencies; the frequency separating these two regimes is denoted the corner frequency, $f_c$. The scaling region with frequencies above $f_c$ carries information about the material properties of the matrix, which includes viscous and elastic properties[8,27,34–36]. At frequencies above the corner frequency, the power spectra, $P_x(f)$ (Fig. 2c), scale with frequency, $f$:

$$P_x(f) \equiv \left\langle \left| \tilde{x}(f) \right|^2 \right\rangle = k \cdot f^{-(1+\alpha)} \quad (1)$$

The scaling exponent, $\alpha$, characterizes the scaling of the mean squared displacement (MSD) of the nanoparticle as a function of time, $t$: $\mathrm{MSD} = \left\langle \left| x(t) \right|^2 \right\rangle \propto t^{\alpha}$[8,34,35]. The scaling exponent provides information about the motion of the tracer: $\alpha = 0$ signifies complete confinement, $0 < \alpha < 1$ indicates subdiffusion in a viscoelastic medium, and $\alpha = 1$ is a sign of

Brownian motion in a purely viscous medium, such as water. When comparing values of scaling exponents, $\alpha$, in the subdiffusive regime ($0 < \alpha < 1$), and if probing at timescales where nonequilibrium processes are negligible, a higher value of $\alpha$ indicates a more viscous environment and a lower $\alpha$ value indicates a more elastic environment. The constant $k$ from Eq. (1) contains information about the signal-to-noise ratio (Supplementary Fig. 1) and does not enter the calculation of $\alpha$.

It is important to note that using Eq. (1) to analyze spectra for the purpose of retrieving $\alpha$ does not require a classical calibration of the optical trap. In scaling analysis it is, however, important to ensure that voltage output from the photodiode is indeed linearly related to the lateral displacements of the tracer particle[37].

In biological systems, nonequilibrium processes have been shown to occur at frequencies below 100 Hz[38–40]. Also, it has been shown that the fluctuation-dissipation is satisfied for both active and passive measurements at frequencies higher than ~10 Hz[41–44]. Beside activities inside cells, other types of noise can occur in the low-frequency region (<300 Hz), for instance from mechanical drifts in the experimental setup. Another point to consider when choosing the frequency region relevant for analysis, is the filtering effect from the photodiode. This can be characterized by a 3 dB cutoff frequency, $f_{3dB}$, which has been measured and reported for our system to occur above 6 kHz[45]. For these reasons, we chose to fit the power spectra in a frequency interval between 400–4000 Hz to obtain $\alpha$.

A typical power spectrum resulting from optical trapping of a microinjected nanoparticle at a depth of about 150 μm in a living zebrafish, as well as a power spectrum obtained by trapping a similar nanoparticle in water are shown in Fig. 2b. As expected, measurements returned $\alpha \approx 1$ for a bead in water[32], whereas the motion of the tracer was subdiffusive within embryonic tissues (with $\alpha = 0.77$ for the experiment depicted in Fig. 2b)[8,36].

To determine the optimal duration of a measurement maximizing the signal-to-noise ratio, we performed an Allan Variance analysis of the equipment, which returned an optimal measurement duration of 2–3 s[46]. In aqueous samples, laser trapping has been reported to heat up the site of the trapped particle by about 1 °C/100 mW[47,48]. As our measurements take place deep into highly scattering tissue, the laser intensity reaching the tracer will be substantially lower than 300 mW, hence, the temperature increase will be less than 2–3 °C. This, in combination with the fact that each measurement lasts 2–3 s renders it likely that temperature effects are negligible.

A classical way of characterizing and comparing viscoelastic properties of different materials independent of the microrheological technique employed for the measurement is by calculating the complex shear modulus, $G(f)$[31,49–52]. The complex shear modulus is the ratio of the shear stress to the shear strain, describing the viscoelastic response of a system to time-dependent stress. It is defined as

$$G(f) = G'(f) + iG''(f) = \frac{1}{6\pi r \gamma(f)} \quad (2)$$

where $r$ is the radius of the bead, $G'(f)$ denotes the storage modulus and describes the elastic response of the system, and $G''(f)$ denotes the loss modulus and describes the viscous response of the system. $\gamma(f)$ is the complex response function, and more information on how $G(f)$ is calculated following the methods outlined in ref. [53] is given in the "Methods" and in Supporting Information.

Importantly, the storage and loss moduli scale with frequency by the same exponent describing the scaling of the power spectral data: $G' \sim f^\alpha$ and $G'' \sim f^\alpha$[31]. While calculating the complex moduli one should be aware of the inherent frequency limitation originating from the finite maximum measurement frequency employed (detailed in Materials and Methods in the Supporting Information).

As a control, we first calculated the complex shear modulus, $G(f)$, for a nanoparticle optically trapped in water. The results for a nanoparticle trapped in water (Supplementary Fig. 3) show the expected behavior of both the loss and storage moduli in the relevant frequency intervals; in particular, $G''(f)$ scales with $\alpha = 1$ as expected for a purely viscous medium where $G = G'' = 2\pi\eta f$. To test the feasibility of the method for characterizing cellular viscoelastic properties of internal tissues, we calculated the scaling exponents, $\alpha$, as well as the storage, $G'(f)$, and loss, $G''(f)$, moduli for a subset of gut and liver progenitors ($n = 5$). Both cell populations have a common endodermal origin and are located about 90–120 μm within zebrafish embryos at 30 hpf. Power spectra obtained from liver and gut progenitors scale with frequency in the interval 400 Hz $< f < 4000$ Hz with distinct scaling exponents ($\alpha_{gut} = 0.56 \pm 0.08$ and $\alpha_{liver} = 0.76 \pm 0.06$ for the depicted experiments), thus inferring differential viscoelastic properties for liver and gut progenitors (Fig. 3a). Calculating the complex shear modulus within the same frequency window and of the same type of data, as analyzed by power spectral analysis, shows that the loss modulus, $G''(f)$, scales with frequency by exponents of $\alpha = 0.53 \pm 04$ and $\alpha = 0.74 \pm 0.05$ for gut and liver progenitors, respectively (Fig. 2d). As expected, these $\alpha$-values are consistent with those obtained by power spectral analysis.

Moreover, trapping beads at 20 and 100 μm depth in Matrigel, representing a material with uniform mechanical properties, returned consistent $\alpha$-values at both depths, demonstrating that the $\alpha$-value does not depend on the depth of the NIR laser trapped beads (Supplementary Fig. 4). Altogether, these data demonstrate that viscoelastic cell properties in deep embryonic tissues can be quantified in vivo based on thermal fluctuation of optically trapped nanoparticles.

**Cell populations within the foregut region display distinct viscoelastic properties.** To quantify the viscoelastic properties of cell populations relevant for foregut morphogenesis, we laser-tracked thermal fluctuations of nanoparticles in the cytoplasm of cells within the developing gut, liver, yolk, and left and right LPM in living zebrafish embryos. After the in vivo measurement, the zebrafish embryo was fixed and imaged by confocal microscopy (representative images are shown in Fig. 1f and Supplementary Fig. 5). In most experiments, the tracer particle was located in the cytoplasm at a distance from the nucleus, and from the actin cortex, that was substantially larger than the amplitude of its thermal fluctuation (50 nm). A typical location of the particle is shown in Supplementary Fig. 5 and quantification of all data is shown in Supplementary Fig. 2).

By coupling the microrheological measurements of each nanoparticle to its location within the specific cell population, using the scaling exponent $\alpha$ as a measure, we generated a spatially resolved viscoelasticity map of the foregut region. This revealed significant differences between both liver and gut progenitor populations and niche tissues directly adjacent to the liver, including parts of the left and right LPM and the yolk (Fig. 3 and Supplementary Fig. 6). The cell-type-specific $\alpha$-values between individual embryos are highly consistent, corroborating the robustness of the approach (Supplementary Fig. 6).

Despite the common origin from the foregut endoderm, gut progenitors stand out as significantly more elastic compared to liver progenitors with average $\alpha$–values of $0.57 \pm 0.07$ ($n = 42$) and $0.66 \pm 0.06$ ($n = 90$), respectively (Fig. 3c). Both in the liver and gut regions, the loss modulus dominates the storage modulus ($G'' > G'$) in the 400–4000 Hz frequency range (Fig. 2d). Also, gut progenitors have a higher storage modulus than liver progenitors. This, together with the significant difference in the $\alpha$-values between the two populations, demonstrates that progenitors in the liver bud are more viscous than gut progenitors, and gut progenitors are more elastic than liver progenitors.

In addition to this relative comparison of viscoelastic properties of the investigated cell populations, the obtained scaling exponents can be quantitatively compared to those previously obtained in a study of viscoelastic properties of gels, specifically Matrigel, with systematically varying polymer concentration ref. [27]. The range of the scaling exponents determined in the current study, $\alpha = 0.53$–$0.74$, are comparable to that determined for Matrigel polymer concentrations between 25 and 50%, thereby providing information on the relative fold change of elastic biopolymer density between the cell populations.

In the surrounding niche, the adjacent LPM epithelia exhibit similar viscoelasticity as liver progenitors ($\alpha = 0.68 \pm 0.06$ and $\alpha = 0.64 \pm 0.06$), while the yolk is more viscous than any of the other tissues ($\alpha = 0.73 \pm 0.08$) (Fig. 3c, d). Moreover, we identified significantly different $\alpha$ values for the left and right LPM, which share the same embryonic origin and epithelial organization, though move asymmetrically during liver morphogenesis (Figs. 1c and 3).

To elucidate how the viscoelastic properties of liver and gut progenitors may arise, we assessed the viscoelastic properties of their endodermal progenitors around 22–23 hpf. The cells

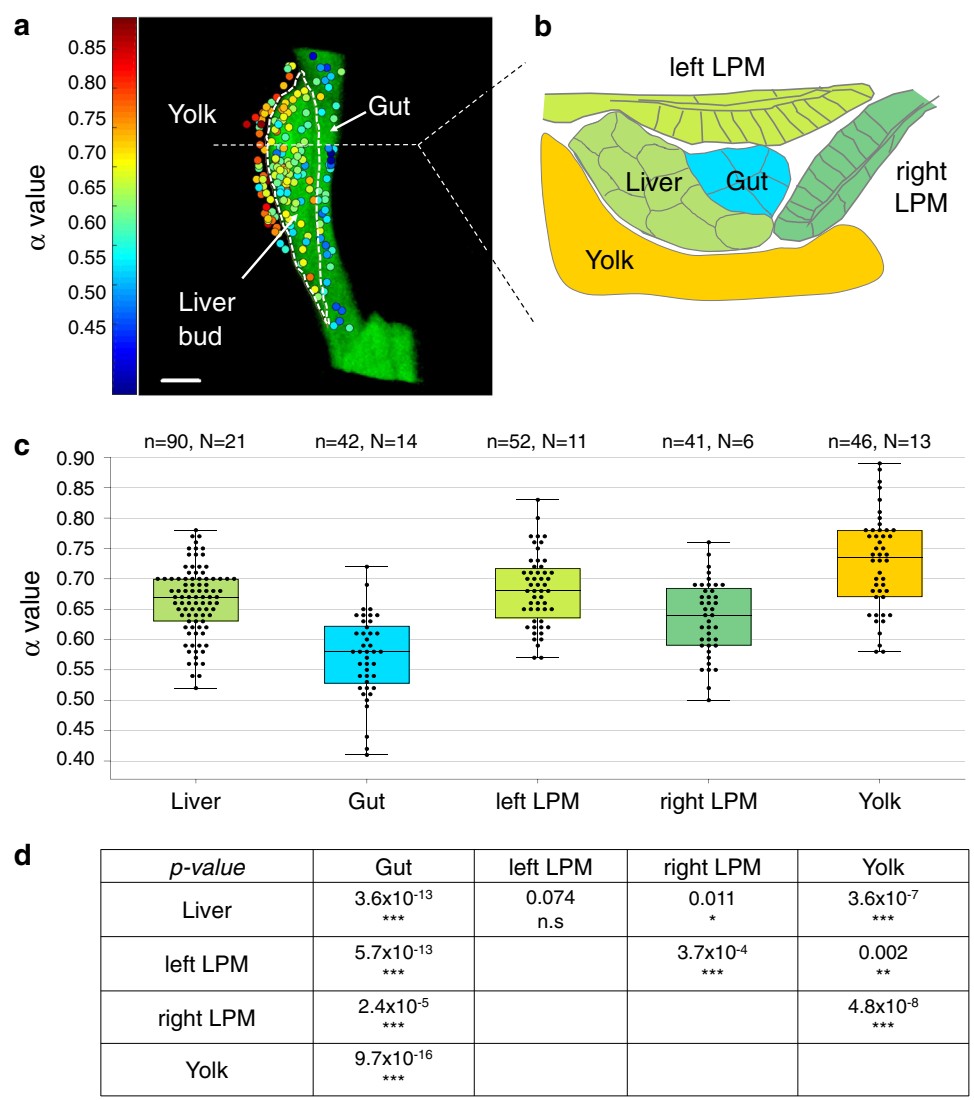

**Fig. 3 Viscoelastic mapping of the foregut region reveals different biomechanical properties of neighboring progenitor populations. a** 2D projection of viscoelasticity map of the liver, gut, and yolk. The color scale on the left indicates cellular viscoelasticity: increasing blue values show more elastic tissues, while increasing red values represent more viscous tissues ($n = 178$ nanoparticles, $N = 32$ embryos). Scale bar: 20 μm. **b** Viscoelasticity map of the foregut region in a projection orthogonal to (**a**) and using color code corresponding to average $\alpha$ values for each tissue type, color scale bar as in (**a**).
**c** Quantification of $\alpha$ values for different tissues shown in a box plot; $n =$ number of analyzed particles, $N =$ number of embryos. **d** Statistical comparison of $\alpha$ value distributions between different tissues. The table provides the $P$ value calculated for each pair of tissues using a two-tailed equal variance Student's *t t*est. *$P < 0.05$, **$P < 0.01$, ***$P < 0.001$.

undergo substantial rearrangement at this stage. This low-frequency movement does not impact microrheology measurements in the 400–4000 Hz frequency range, however, it does complicate optical trapping and particle localization. Nevertheless, we trapped several injected particles at 22–23 hpf and the obtained $\alpha$-value ($0.62 \pm 0.06$) falls between the $\alpha$-values for liver ($0.66 \pm 0.06$) and gut ($0.57 \pm 0.07$) progenitors about 6 h later (Supplementary Fig. 7). These data show that cell populations have characteristic viscoelastic properties, which may change as they adopt specific fates.

**Microtubule concentrations are higher in gut than liver progenitors**. Given the cytoplasmic viscoelastic properties of gut progenitors differ significantly from those of liver progenitors in a frequency interval relevant for cytoskeletal biopolymer dynamics[22], we next investigated which biopolymers could be responsible for this observation. We analyzed the distribution of actin filaments and microtubules, as they are major contributors

to cytoskeletal stiffness and dynamics. For both tissues, we found that actin localizes mainly to the cell cortex, while the microtubule network is distributed throughout the cytoplasm excluding the volume of the cell nucleus (Fig. 4a–c). Notably, microinjected nanoparticles are surrounded by the microtubule network and typically do not co-localize with actin filaments (Fig. 4b, c and Supplementary Fig. 5). For both liver and gut progenitors, most nanoparticles are further away from the actin cortex and from the nucleus than the typical amplitude of their thermal fluctuations within the optical trap (~50 nm; Supplementary Fig. 2). This makes the microtubule network the prime molecular candidate to mediate the different viscoelastic properties between progenitor populations. High cell density and complex 3D arrangement of cells within the foregut region make it difficult to analyze the microtubule cytoskeleton at the single-cell level. Therefore, we quantified average microtubule density for gut and liver progenitors on the tissue level. To avoid region-specific and cell orientation bias, analyses were performed on volumes with linear

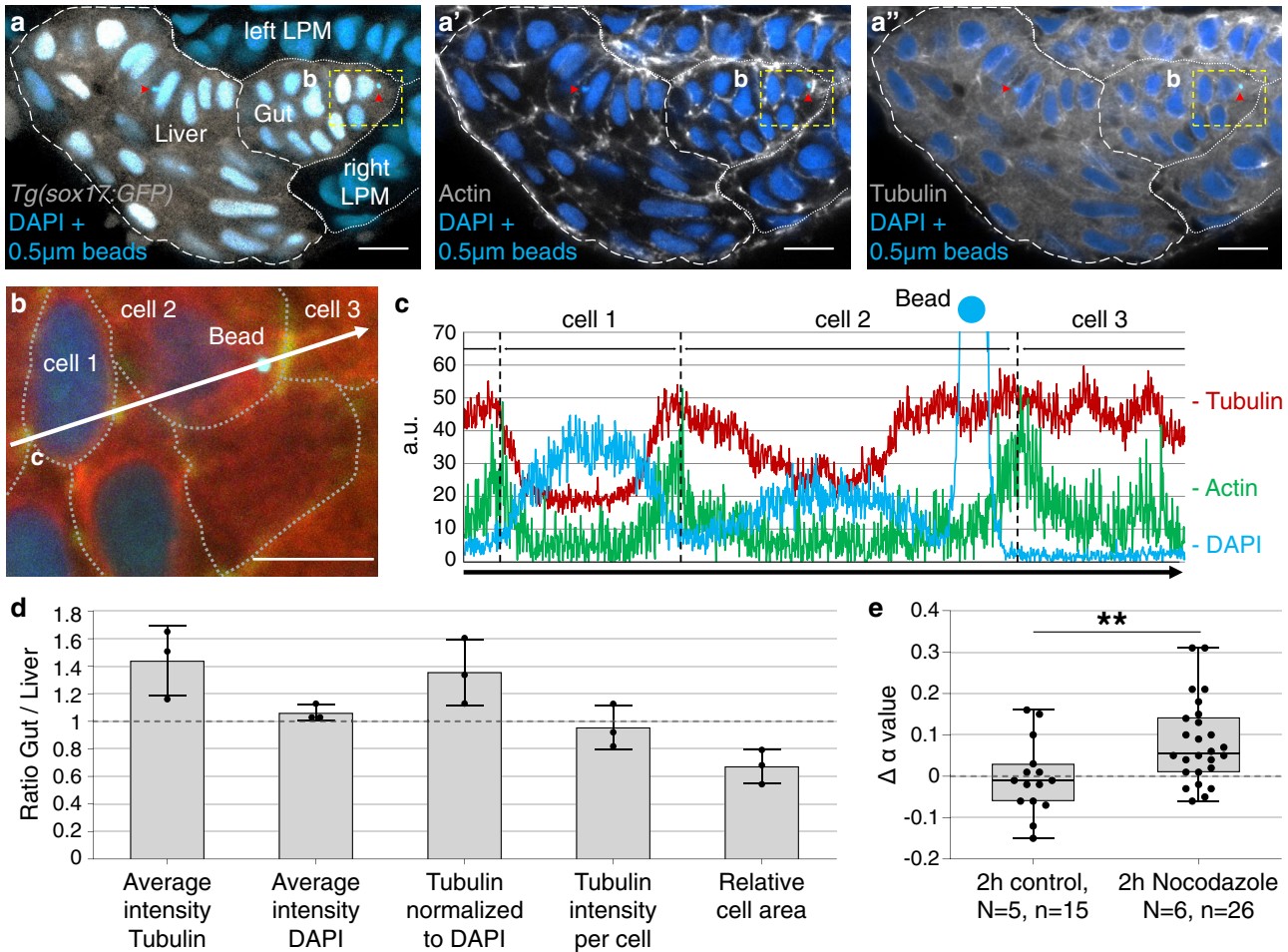

**Fig. 4 Microtubules surround injected nanoparticles and their concentration is increased in gut progenitors. a–a″** Transverse section of a 30 hpf *tg(sox17:GFP)* embryo microinjected with 0.5 μm fluorescent polystyrene beads (light blue dots next to red arrowheads) showing Phalloidin staining of actin filaments (gray in (**a′**)) and β-tubulin staining to visualize microtubules (gray in (**a″**)). Fluorescent emission of microinjected beads and nuclear DAPI occurs at similar wavelengths; beads are detected by size and high signal intensity. Scale bar: 10 μm. **b** Magnification of cellular bead location (yellow dashed rectangle in (**a–a″**)). Cell borders are outlined based on the presence of cortical actin (dashed lines). Scale bar: 20 μm. **c** Intensity profile of actin, tubulin, and DAPI along the white arrow in (**b**). The high peak in the DAPI channel indicates bead position. **d** Ratio of tubulin levels between gut and liver determined from tissue volumes with a typical linear dimension of 20 μm (see "Methods" and Supplementary Fig. 5). The average intensity represents the signal intensity normalized to tissue area (tubulin) or nuclear area (DAPI) for each optical section. Tubulin intensity per cell is calculated by normalizing the average Tubulin intensity to the nuclei number. The relative cell area was obtained by normalizing the volume of the respective tissue to nuclei number. Error bars represent one standard deviation, number of analyzed embryos = 3. **e** α-value changes after 2 h of 2 μM Nocodazole treatment (green; mean +SD = 0.082 ± 0.1) and in DMSO controls (blue; mean+SD = −0.004 ± 0.09). N = number of embryos, n = number of measured nanoparticles; P value calculated using unpaired equal variance two-tailed t test; **P = 0.0086.

dimensions of 20 μm, yielding cumulative values from individual optical sections. The relative amount of microtubule is 1.44 ± 0.25-fold higher in the gut compared to liver progenitors (Fig. 4d and Supplementary Fig. 5e). The intensity of nuclear DAPI of the respective tissues was used to correct for a potential bias during data acquisition. This revealed that the amount of microtubule per cell is similar between the two populations (0.96 ± 0.16-fold). However, the area of gut progenitors is only 0.67 ± 0.13-fold of that of liver progenitors. Hence, despite a similar cellular amount of microtubules in the two populations, the microtubule concentration within gut progenitors is about 1.36 ± 0.24-fold higher per cell.

To assess whether the microtubule cytoskeleton influences viscoelastic cell properties, we disrupted microtubule polymerization using drug treatments with 2 μM Nocodazole. We measured α-values for the same nanoparticles located in gut progenitors before (at 28 hpf) and after 2 h of drug treatment. In controls, particles were measured at the same 2-h interval without drug

administration. Microtubule destabilization leads to a significant increase in the α-value indicating a shift to more viscous cell properties (Fig. 4e and Supplementary Fig. 8a, b). Concomitantly, microtubule destabilization during foregut morphogenesis impairs asymmetric gut looping and liver bud formation (Supplementary Fig. 8c–f). Thus, the microtubule cytoskeleton influences viscoelastic cell properties and microtubule concentration correlates with the differential elasticity between gut and liver progenitors.

## Discussion

We performed nearly non-invasive quantitative microrheological investigation of cells and tissues at depths up to 150 μm in a living embryo. Using the cell populations within the developing foregut region as a model to investigate material properties in vivo, we show that cells exhibit consistent viscoelasticity within a population, whereas viscoelasticity significantly differs between various cell types, with gut progenitors being more elastic than any other

population in the region. On the molecular level, we show that the more elastic property of gut progenitors correlates with a higher concentration of microtubules in comparison to the liver progenitors and that microtubule disruption alters cellular viscoelasticity, as well as foregut morphogenesis.

We tracked the thermal fluctuation of a tracer particle using a QPD which picked up the forward-scattered light, a method which was previously applied to study transport mechanisms and material properties of single cells[8,30]. We used a sampling frequency of 22 kHz, as this is the frequency minimizing the Allan Variance of our setup[46]. However, the quadrant photodiode used for data acquisition can sample with frequencies up to 100 MHz. Hence, even without pushing the limits of the quadrant photodiode, this type of data acquisition is several orders of magnitude faster than previously published particle tracking methods based on video recording[16,19,22,54,55], thereby allowing for a substantial expansion of the timescale from which information can be retrieved.

The intracellular tracer nanoparticles used in this study are found neither directly adjacent to the cell cortex nor to the nucleus, thus, they mainly provide information about the viscoelastic properties of the cytoplasm. For both liver and gut, we find $G'' > G'$ at all frequencies, hence, for the intracellular environment in the developing gut region the viscous properties dominate over the elastic. Our data for cells located in internal tissues 90–150 μm deep within zebrafish embryos agree with video-based tracer measurements inside the *Drosophila* syncytium[22] which were performed at depths up to 40 μm in the frequency interval $0 < f < 1000$ Hz. In accordance with our findings, the cytoplasmic viscoelastic properties in *Drosophila* embryos were shown to be dependent on microtubules rather than the actin cytoskeleton. In contrast, atomic force microscopy (AFM)-based microrheology performed on isolated cells report $G' > G''$ for similar frequencies[56–58]. This is likely due to the extracellular location of the probe as opposed to our method; an AFM operates on the cell surface and cortical actomyosin therefore notably contributes to such microrheological measurements (Brückner et al.[58]; Rigato et al.[56]).

This study was not targeted at measuring absolute forces nor distances inside the zebrafish. If this had been the target, one could potentially use oscillatory optical tweezers for this, even at frequencies where nonequilibrium processes contribute to the signal[59,60], although the crowdedness around the tracer particle would complicate this.

AFM and techniques such as microfluidics-based assays[61] or optical stretching[62] can measure the viscoelasticity of isolated cells and tissues, however, they cannot be directly used for internal tissues in vivo. Other methods described to investigate biomechanical properties in deep tissues, including Brillouin microscopy or tomography[63,64], report on the length scale of tissues. In contrast, the presented laser-based assay causes no detectable physiological damage and can be used to probe single cells in intact tissues and embryos without disrupting their native environment. By laser-tracking intracellular tracer nanoparticles, we demonstrate that different cell populations have distinct viscoelastic properties indicating that cytoplasmic viscoelasticity is cell-type specific and can be used to distinguish cell populations in vivo.

In our study we used a 1064 nm NIR laser to detect thermal fluctuations of nanoparticles within zebrafish embryos that have a width of ~300 μm. The penetration depth of NIR lasers into biological tissues depends, e.g., on laser power, the setup, and tissue scattering, but tens of mm are achievable. Due to the high penetration of NIR light, the optical trapping method described here should be applicable to any organism or tissue amenable to NIR, and therefore should be suitable for probing biomechanical

properties in vivo in species other than zebrafish, as well as complex in vitro cultures, such as explants and organoids. The major challenge could be the visualization of the particles within deep tissues. In our study, during the optical trapping experiments, we used standard confocal microscopy with a low intensity 405 nm laser to locate single polystyrene beads at depths up to 150 μm. The resolution at this depth was relatively poor and we did not push to improve it as we wanted to minimize photodamage of the living zebrafish. The precise location of the nanoparticles was more carefully examined after the trapping experiments by high-intensity and high-resolution (~250 nm) confocal microscopy using fixed and deyolked embryos. One improvement of the setup could be to use microscopy techniques with higher resolution during the optical trapping experiments, such as multiphoton[65,66], Airy beam imaging[67], or by fluorescent label optimization.

The optically tracked nanoparticle returns information about the local viscoelastic environment inside living cells. The application of this technique could be straightforwardly expanded to examine tracers located in the extracellular space, for instance in the extracellular matrix, which is of interest due to its importance for the differentiation of embryonic and induced stem cells[68]. In addition, this method can be adapted for active microrheological and force measurements[20], as optical trapping allows manipulation of the particle location.

The timescales here investigated are those relevant for cytoskeletal dynamics, including microtubule turnover and polymerization. Recent work shows cell-autonomous functions for microtubule-mediated mechanics in the developing *Drosophila* wing epithelium, thus providing evidence for the contribution of material properties to tissue scale morphogenesis[69]. At the stage of liver bud formation, gut progenitors exhibit little to no motility and shortly after give rise to the intestinal epithelium. At the same time, however, both liver progenitors and LPM undergo dramatic cellular rearrangement before differentiating into their respective cell types. Although a more systematic analysis of the cell property changes occurring over the course of development is needed, our results show that such changes parallel fate differentiation, suggesting they may drive underlying cell behaviors and/or tissue differentiation. The comparatively fluid nature of the liver progenitors and LPM epithelia corroborates their active migration and movement[24,25]. In contrast, higher tubulin concentration of the more static gut progenitors, as found in the current study, is consistent with an increased stiffness to counteract the dynamic rearrangement of surrounding tissues, given that microtubules are the stiffest cytoskeletal filaments[70]. Consistently, we find that microtubule disruption results in decreased gut progenitor elasticity and altered foregut morphology. Furthermore, our results in conjunction with lower cell rearrangement among gut than liver progenitors[24], are comparable to cell behaviors associated with tissue stiffness during axis elongation in zebrafish[71]. A similar relationship between motility and viscoelasticity has also been observed in cancer spheroids, where the motile cells located at the invasive tips of the spheroid are more viscous than those statically located at the base which are more elastic[2]. In our work we find the yolk adjacent to the forming foregut to be the most viscous region probed, indicating an environment that is permissive to dynamic tissue rearrangements. Altogether these results suggest that differences in material properties between adjacent developing tissues may drive or facilitate cell movement and/or tissue differentiation.

In summary, we present a method for quantifying and mapping material properties in cells and tissues in vivo, which is a prerequisite for establishing a firm connection between material properties, biomechanics, and cell behaviors in development and disease.

## Methods

**Experimental model and subject details.** Adult zebrafish and embryos were raised according to standard laboratory conditions[72], and all experiments were performed in agreement with the ethical guidelines approved by the Danish Animal Experiments Inspectorate (Dyreforsøgstilsynet). The transgenic line $Tg(-0.5\ sox17:GFP)^{zf99}$ was used to visualize the endoderm[73]. To prevent pigment formation, after 24 h hpf embryo medium (120 mg/L sea salt; Instant Ocean, Aquarium Systems, France) was supplemented with 0.2 mM PTU (1-phenyl-2-thiourea; Sigma-Aldrich, USA).

**Preparation of zebrafish embryos.** 0.5-μm polystyrene fluorescent beads (FP-0545-2; SPHERO Fluorescent Particles, Light Yellow, 0.4–0.6 μm; Spherotech, USA) were injected at the 1-cell stage into the cytoplasm–yolk interface. Prior to injections, the stock solution of fluorescent particles (1% w/v in deionized water with 0.01% NP40 and 0.02% Sodium Azide) was diluted 1:10 in autoclaved deionized water, and 0.5-1 nl was injected per embryo to ensure a sufficient number of beads in the foregut region at 28–30 hpf and reduced extracellular clusters. At 28–30 hpf, injected embryos were manually dechorionated and anesthetized by incubation with 0.14 mg/ml Tricaine (ethyl 3-aminobenzoate methanesulfonate; A5040, Sigma-Aldrich, USA).

Embryos were embedded dorsolaterally on the left side, with the liver bud oriented towards the coverslip (25 × 60 mm #1 coverslip, Menzel-Gläser, Germany), in a drop of 0.4% low melting temperature agarose (NuSieve GTG Agarose; Lonza, USA). Before agarose polymerization, embedded embryos were covered with another coverslip (24 × 40 mm #1; Menzel-Gläser, Germany). The two coverslips were separated by vacuum grease (Dow Corning High Vacuum Grease) applied on the longitudinal edges of the second coverslip. After agarose polymerization, the remaining space in the chamber between the coverslips was filled with approximately 200 μl embryo medium containing PTU and Tricaine, as described above. Finally, the chamber was sealed by vacuum grease.

**Optical trapping and confocal imaging.** A near-infrared laser beam ($\lambda = 1064$ nm, Nd:YVO4, Spectra- Physics J20-BL106Q) was implemented into a Leica SP5 confocal microscope (Fig. 2a), thus allowing for simultaneous optical trapping and visualization; for details of the setup see reference[29]. A Leica (PL APO, NA = 1.2, ×63) water immersion objective was used for focusing the trapping laser beam inside the sample as well as for confocal image acquisition. Forward-scattered laser light was collected by an immersion oil condenser, located above the sample, and imaged onto a Si-PIN quadrant photodiode (Hamamatsu S5981) placed in the back focal plane. Data were collected using custom-made Labview programs using a sampling frequency of 22 kHz.

In parallel with optical trapping of the nanoparticles, confocal imaging was performed using excitation lasers with wavelengths of 405 nm and 488 nm, thereby exciting fluorophores on the nanoparticles and visualizing the foregut endoderm. For all measurements, the laser was operated for 2–3 s, thus minimizing the Allan Variance of the setup[46] and thereby maximizing the signal-to-noise ratio, and using the power of ~300 mW in the sample. In agreement with the literature[14], no physiological damage was observed as a consequence of this irradiation. Also, the expected temperature increase during the 2–3 s measurement interval is well below 2–3 °C[48]. All measurements were performed at room temperature.

During measurements, we visually in three dimensions co-localized the focus of the optical trap with a single nanoparticle inside the fish. Importantly, the amplitude of the power spectrum substantially increased upon nanoparticle trapping compared to having an empty trap inside the fish (Supplementary Fig. 1), because upon correct alignment of the optical trapping system, a nanoparticle in the focus of the trap serves as a lens increasing the number of photons reaching the QPD. Using a 3D piezo stage (Mad City Labs) the position of the optical trap was fine-tuned in order to maximize the amplitude of the power spectrum and the measurement was acquired with these settings.

**Localization of injected nanoparticles.** To analyze overall nanoparticle distribution, cell membranes were mosaically labeled by co-injecting Tol2-Ubi-mKate-CAAX plasmid DNA (courtesy of Sara Caviglia). Liver progenitors were visualized by the immunostaining for EfnB1[24].

To determine the precise location of nanoparticles used for measuring viscoelasticity, the embryos were fixed in 4% paraformaldehyde (PFA) immediately after the optical trapping experiments, deyolked, and stained with phalloidin conjugated to Atto 633 (Sigma-Aldrich, USA) to visualize the actin cell cortex and tissue morphology. The foregut region was imaged using a Leica SP8 confocal microscope. Images obtained from confocal scanning simultaneously with the trapping experiments were used to identify the particles of interest in the fixed sample by manually correlating foregut shape and nanoparticle distribution in 3D using Bitplane IMARIS software. The nanoparticles were assigned to a specific cell population based on tissue morphology and transgenic sox17:GFP expression. Only single beads, which could be clearly assigned to a specific tissue, were included in the analysis.

**Visualization of microtubules and actin filaments.** Immunostaining for β-tubulin was used to visualize microtubules, while actin filaments were stained using Phalloidin-Atto 633 fluorophore (Sigma-Aldrich, St. Louis, USA). Immunostaining was performed

on sections to reduce penetration bias. Embryos were fixed at 30 hpf in 4% PFA overnight at 4 °C, washed in 1× PBS (Gibco, Life technologies), deyolked, and embedded in 4% agarose (Ultra Pure Agarose, Invitrogen) for subsequent sectioning. Transverse 50-μm sections were obtained using a Leica microtome and stained with mouse monoclonal β-tubulin antibody (1:50; E7, DSHB; 1–2 μg/ml) followed by secondary goat anti-mouse Cy3 antibody (1:300; Jackson ImmunoResearch). To visualize actin filaments in embryos microinjected with 0.5-μm nanoparticles, sections were co-stained with phalloidin-Atto 633 (1:300) and DAPI (1:1000). To discriminate liver from gut progenitors for quantification of microtubule density, embryos were co-stained with either rabbit polyclonal EfnB1 (details in ref. [24]; 1:1000) or Prox1 (1:500; AngioBio) antibody, followed by goat anti-rabbit Alexa 633 (Jackson ImmunoResearch; 1:300). For confocal imaging sections were mounted in VECTASHIELD Antifade Mounting Medium (Vector Laboratories, USA) on the glass slides (Superfrost Plus; Menzel-Gläser, Germany) and covered with #1 coverslip (Menzel-Gläser, Germany). Consecutive optical sections were collected every 0.33 μm using a Leica SP8 confocal microscope.

**Drug treatment for microtubule depolymerization.** Embryos were embedded as described above and nanoparticles in gut progenitors were used for measurements at 28 hpf. After the measurements, incubation medium in the chamber was replaced with about 200 μl embryo medium containing 2 μM Nocodazole (Sigma-Aldrich, USA), 0.2 mM PTU, and 0.14 mg/ml Tricaine. After 2 h of incubation at room temperature (average 24 °C), measurements were performed on the same nanoparticles. Embryos embedded in agarose remained in the same position during the medium exchange, which facilitated the localization of previously measured nanoparticles. For controls, the same nanoparticles were measured at 28 hpf and 2 h later without Nocodazole incubation.

**Probing viscoelasticity at different depths in a uniform viscoelastic material.** In all, 1-μm beads were optically trapped at different depths inside a DMEM: Matrigel (in a ratio of 3:1) viscoelastic matrix, details of the experiment are given in ref. [27]. To prepare the matrix, frozen matrigel was slowly thawed on ice and mixed with cooled-down DMEM and nanoparticle solution. The matrix was injected between two coverslips and the chamber was sealed with vacuum grease. The chamber was placed on the microscope stage and microbeads were trapped at depths of 20 and 100 μm inside the matrix. The obtained measurements were analyzed as described below and the data are shown in Supplementary Fig. 4.

**Statistics and reproducibility.** All data are presented as mean ± s.d. Statistical significance was calculated using a two-tailed equal variance Student's $t$ test, for the exception of Supplementary Fig. 8 where a two-tailed paired $t$ test was used. Exact $P$ values are indicated in the figure legends and Fig. 3d, with the following abbreviations being used for the figures: n.s. $P \geq 0.05$, $*P < 0.05$, $**P < 0.01$, $***P < 0.001$. The sample size is specified in the corresponding figures, and/or described in the figure legends. Unless otherwise specified, $n$ refers to individual replicates (e.g., individual beads), and $N$ refers to the biological replicates (e.g., individual embryos). All data used for the statistical analysis can be found in the Supplementary Information.

**Reporting summary.** Further information on research design is available in the Nature Research Reporting Summary linked to this article.

## Data availability

The datasets generated during and/or analyzed during the current study are available from the corresponding authors on reasonable request. Source Data are available in Supplementary Data 1.

## Code availability

All computer code and software used in the current study are listed in the supporting information, are commercially available, and have been previously reported.

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

## Acknowledgements

We thank Charlotte Bailey, Sara Caviglia, Jakub Sedzinski, and Raghavan Thiagarajan for their helpful comments on the manuscript and S.C. for *ubi:CAAX-mKate* plasmid. This work is supported by the Danish National Research Foundation via the StemPhys CoE (DNRF116) and by the Novo Nordisk Foundation to E.A.O. (NNF17CC0027852).

## Author contributions

E.A.O. and L.B.O. conceived, designed, and supervised the project. A.D. and Y.F.B. performed all experiments and analyzed the data. All authors discussed the results and implications and wrote the manuscript.

## Competing interests

The authors declare no competing interests.
