## [Peer Review File · Communications Biology]

Reviewers' comments:

Reviewer #1 (Remarks to the Author):

The paper entitled "Foregut organ progenitors and their niche display distinct viscoelastic properties in vivo during early morphogenesis stages" authored by Dzementsei, Barooji, Ober and Lene B. Oddershede is using laser tracking and photodiode detection of thermal fluctuations of optically trapped nano and micro-particles at rather large depths in Zebrafish to study the differences in viscoelastic properties between progenitor populations. The particles of gold and labelled polystyrene are microinjected into the Zebrafish for this study. This is done in order to study mechanical properties of these developing organs and the differences in this behaviour are observed. The study is done using optical tweezers in combination with confocal microscopy. In the study of the viscoelastic properties of the foregut region it is shown that progenitors in the liver bud are more viscous than gut progenitors and gut progenitors are more elastic than liver progenitors. The results are convincing and promising. The authors also say that these results suggest that differences in material properties between adjacent developing tissues may facilitate cell movement and/or tissue differentiation. I agree that it might be so however I have a couple of questions. Firstly, it is my feeling that the authors are not doing quantitative studies of the behaviour but rather only a relative measurement is done. They do give the values for the viscosities and elasticities and the differences can be observed but I am not sure with relation to what. It is true that they are different but why shouldn't they be. I also think that these values are established under rather strict conditions that are not stated in the paper. So, one of them would be that all the analysis assume that the system can be described as a harmonic oscillator and that all the measurements are done in the linear regime but these conditions are not necessarily fulfilled in the experiments. In order to do proper quantitative measurements several conditions have to be specified and they are not. Some of the measurements are done on the microscope slide and their relation to the in vivo measurements is not clear. The other thing is that I cannot see how the calibration of the whole system is done. Does it rely on separate calibration or is the calibration done in situ? Or the calibration for some reason is not needed.

On the minor side of comments: The claim in the introduction that these are the deepest measurements into the living tissue are not exactly right. Firstly the paper J. Opt.15(2013) 075305 (8pp) demonstrates methods for trapping into depth of up to 166 micrometres. The authors also describe the use of adaptive optics in an optical trapping system for enhanced lateral trap stiffness at depth. Secondly, in the Nature Comm paper DOI: 10.1038/s41467-017-00713-2, the authors describe trapping into 150 micrometres depth in live Zebrafish. So, I would suggest softening of the statement in the introduction about depth trapping. I think that it would be sufficient to say what the depth is even if others have done other measurements at this depth.

Fig S3 is done under different conditions than the subsequent measurements in the Zebrafish.

Reviewer #2 (Remarks to the Author):

The manuscript describes the experimental method and the result of optical trap-based measurement of viscoelasticity in deep tissues. The authors demonstrated significant α value differences between the gut and liver, showing the method's promise.

Here are some comments to be addressed.

(1) The abstract should include the quantified α values and a brief summary of the discussion/conclusion on the measured values.

(2) "we measure material properties at sub-cellular resolution in deep tissues using laser tracking and photodiode detection of the thermal fluctuations of optically trapped nanoparticles."

"Subsequently, the embryo was fixed and the foregut region was imaged with improved resolution by confocal microscopy to determine the location of each of the tracked particles."

The authors claim that the use of NIR laser allows the analysis of deep tissues. However, the location of the particles was assessed by confocal microscopy, which may negatively impact the

advantage of the NIR laser. Please include a quantitative discussion on the achievable optical resolution with respect to the depth in the zebrafish body for visible and NIR wavelengths.

(3) "We analysed the distribution of actin filaments and microtubules, as they are major contributors to cytoskeletal stiffness and dynamics." "Notably, microinjected nanoparticles are surrounded by the microtubule network but typically do not co-localize with actin filaments".

This means the measured viscoelasticity is always biased by how (where) nanoparticles stay in the tissue/cell. Is there a possible method to assess the contributions from actin filaments and other possible factors than microtubules or to find 'unbiased' viscoelasticity from your measurement?

(4) The measurement was made at 28-30 hpf. How do the α values change at different (earlier) time points? When is the earliest time point to show the α value difference between the gut and the liver?

Reviewer #3 (Remarks to the Author):

Summary:

The authors present an interesting study mapping the viscoelastic properties during early morphogenesis. While this paper may report novel claims that would be of interest to the wider field, it is very difficult to interpret because of significant technical flaws or perhaps lack of detailed explanation. Thus I would like to state that the research question is an interesting one and would have value to the community, but here I will focus on the technical flaws that make the data difficult to interpret. These technical flaws must be resolved before the conclusions of this paper can be interpreted.

Overall, there seem to be two major technical flaws: (1) Parameters (beta and kappa) from optical trap calibration in water seem to be uniformly applied to trapping experiments in-vivo, which will yield incorrect absolute values for any position or force calculations. (2) The assumption of thermal fluctuations within a small range (400-4000 Hz) seems to be applied uniformly over all frequencies, which will yield incorrect material properties. More detailed comments follow below.

Major comments:

- On page 4, the authors refer to the active-passive measurements — This is not just a different approach, but currently is one of the only approaches to get quantitative force measurements in living cells using a QPD detection system. Further, there have been a large number of studies using similar active-passive approaches which results relevant to this study.
- On page 5, the authors state they measure material properties using QPD detection of thermal fluctuations of optically trapped nanoparticles. The main issue in this study is the authors are measuring fluctuations due to all sources, not just thermal forces. Importantly, in a developing cell there are many active internally generated (non-thermal) forces that are contributing to the fluctuations measured. This is the main factor that makes their results of material properties difficult to interpret. For instance, the power-law scaling of the power spectrum cannot be attributed to solely the viscoelasticity of the cells but it is also affected by any activity in the cell. The power-law is thus a combination of both material properties and active fluctuations, not material properties alone.
- On page 6, they state the viscoelastic properties can be mapped by tracking thermal fluctuations. This is again the problem. The fluctuations are not solely thermal. Thus they cannot directly related the measured fluctuations to viscoelastic properties.
- On page 11, the authors state they focus on the frequency regime where non-equilibrium processes are negligible between 400-4000Hz. It would be worth showing that this is the frequency range where non-eq processes are actually negligible. But assuming this is correct, all analysis of material properties must not be limited to this regime. Any frequencies outside this window cannot be analyzed in terms of the fluctuation dissipation theorem. Therefore, the G^* shown in Fig 2d is invalid outside of those frequency ranges.
- Page 13, the authors summarize that their measurements show the viscoelastic properties can be quantified in vivo based on thermal fluctuations of nanoparticles. Again this is the problem - the fluctuations are not thermal overall. And while I agree the fluctuations above a certain frequency

regime may be thermal, that frequency regime must first be verified and then this type of analysis (that relies on thermal fluctuations) cannot be applied outside that frequency range. Further, force calibration must be done in-situ at the time of each measurement since the trap stiffness will depend on the local environment of each cell. Using a constant value for optical trap calibration confounds all results comparing between different cells and locations.

- Page 15, fig 3, shows the non-dimensional alpha parameter. This value is the only quantitative metric I see in the study that allows comparison between different foregut regions because the absolute values of the spectrum do not matter (which are affected by the calibration). The non-dimensional scaling should still be valid. If the goal is to compare material properties, this value can at least accomplish a relative comparison within the region of 400-4000Hz.

- Page 19, the authors state that this is a significant improvement over particle tracking in video recording, and that is certainly true. But it might be a good idea to cite others who have done laser tracking with a QPD in cells previously such as: Yamada, S., Wirtz, D., & Kuo, S. C. (2000). Mechanics of living cells measured by laser tracking microrheology. *Biophysical journal*, 78(4), 1736-1747. And many others.

- SM page 1, it looks as if the authors calibrate their value of β (and subsequently κ) using a bead trapped in water. This is clearly not applicable in their living samples. The values of β and κ will not only differ from in water greatly, they will depend on space and time inside of the living organism. If this is how the calibration was done, then it seems nearly impossible to interpret the absolute values of any of the measurements. If this is not how the calibration was done, the paper needs to be revised extensively to describe how force calibration was carried out.

- SM page 2, equation 4 is only valid in thermal equilibrium since it is based on the fluctuation dissipation theorem. Assuming fluctuations are thermal from 400-4000 Hz, the authors can only use this equation in that range to calculate the imaginary part of the response function. But the authors use this equation to cover 10Hz - 10kHz, which is well outside the range of thermal fluctuations. Since 400-4000Hz limits them to one decade in frequency, it is also questionable if the scaling can be even called a power-law since an exponential decay over 1 decade looks very similar. Further, the G' values will be over an even smaller frequency range (400 - 1000 Hz) due to the Kramers-Kronig relation. If the authors wish they can regain some of their range in G' by using more recent methods: Nishi, K., Kilfoil, M. L., Schmidt, C. F., & MacKintosh, F. C. (2018). A symmetrical method to obtain shear moduli from microrheology. *Soft matter*, 14(19), 3716-3723.

Minor comments:

- The paper is missing reference to the wide body of literature on non-equilibrium fluctuations in cells, which are precisely the studies that show the authors approach does not yield true material properties. I encourage the authors to include some of these citations from the last 5 years which showed an explosion of studies in this area.

- Page 8, Figure 2 is pixelated and difficult to read. Why is G'' omitted for water? Panels B,C, do not show an observable plateau in the low frequency range for particles in the cells — This suggests either that the particles are not trapped or they exhibit active fluctuations in the low-frequency.

- Page 11, they estimate the heating affects based on other studies and state they are negligible. However this is something that can be verified very easily in their own system. The authors should do so, and likely already have the data (repeated measurements in same spot) to check this.

Reviewer #1

We thank Reviewer 1 for carefully reading our manuscript and for providing constructive feedback helping us to improve the manuscript. We were delighted to read that Reviewer 1 found the results to be convincing and promising and hope that, with the current revisions, Reviewer 1 will find his/her remarks answered in a satisfactory manner.

Comment 1-1

Firstly, it is my feeling that the authors are not doing quantitative studies of the behaviour but rather only a relative measurement is done. They do give the values for the viscosities and elasticities and the differences can be observed but I am not sure with relation to what. It is true that they are different but why shouldn't they be.

Answer: It is correct that the presented measurements are relative and show that the gut is more elastic than the surrounding tissues. To our knowledge, these results are the first in literature to quantitatively demonstrate *in vivo* that internal organs have different viscoelastic properties. Our results enable a relative comparison of viscoelastic properties between cell populations which is key for understanding distinct cell and tissues behaviours. Moreover, the range of the obtained scaling exponents can be compared to those obtained in Ref 27 (Borries et al. 2020) where the viscoelastic properties of Matrigel were investigated while systematically varying the polymer concentration. The scaling exponents discovered in the current study, $\alpha=0.53-0.74$, hence correspond to Matrigel polymer concentrations of 25-50 %, providing insights into the fold-change of elastic polymer composition between the respective cell populations. This is more thoroughly discussed in the revised paper.

Comment 1-2

I also think that these values are established under rather strict conditions that are not stated in the paper. So, one of them would be that all the analysis assume that the system can be described as a harmonic oscillator and that all the measurements are done in the linear regime but these conditions are not necessarily fulfilled in the experiments.

Answer: The only assumed condition of the biological system is that we observe a frequency range (400 Hz – 4000 Hz) in which non-equilibrium processes are not dominant. This is in line with observations in the literature, see, e.g., Ref 22 (Wessel et al., 2015). It is correct that for the detection, we assume we are operating within the linear range of the photodiode, i.e., that the voltage signal is linearly correlated with the position of the tracer particle. This is a routine test we always perform on our equipment, as for instance described in Figure 3 of Ott et al., Optics Express vol 22 p 23661, 2014. This paper is cited in the revised manuscript and the point is also discussed in more detail.

Comment 1-3

In order to do proper quantitative measurements several conditions have to be specified and they are not. Some of the measurements are done on the microscope slide and their relation to the in vivo measurements is not clear.

Answer: The optical tweezers are implemented in a confocal microscope. Hence, all measurements performed with the optical tweezers, both *in vivo* (with zebrafish embryos) and *in vitro* (with beads in water and in Matrigel), take place in the setup shown in Figure 2A and described in the “METHODS” section. In case of zebrafish embryos, they are carefully mounted on the slide and immersed in 0.4% agarose. In this condition, the zebrafish embryos are healthy and viable, as they develop normally when released from the agarose after 3-4 hours.

Comment 1-4

The other thing is that I cannot see how the calibration of the whole system is done. Does it rely on separate calibration or is the calibration done in situ? Or the calibration for some reason is not needed.

Answer: One advantage of the proposed method is that there is no need to calibrate the optical tweezers to determine α . In order to determine α , we examine the scaling properties of the power spectrum, or of G' or G'' . For this, we do not require knowledge of for instance the spring constant of the optical trap nor of the conversion factor between voltage and displacement, we just need to know that the latter two are linearly correlated, as discussed in the Answer to Comment 1-2. All conclusions of the paper are solely based on α , therefore, an important aspect of the paper is that a normal optical tweezers calibration procedure is not needed to determine α . Hopefully this is now clarified in the revised version.

Comment 1-5

On the minor side of comments: The claim in the introduction that these are the deepest measurements into the living tissue are not exactly right. Firstly the paper J. Opt.15(2013) 075305 (8pp) demonstrates methods for trapping into depth of up to 166 micrometres. The authors also describe the use of adaptive optics in an optical trapping system for enhanced lateral trap stiffness at depth. Secondly, in the Nature Comm paper DOI: 10.1038/s41467-017-00713-2, the authors describe trapping into 150 micrometres depth in live Zebrafish. So, I would suggest softening of the statement in the introduction about depth trapping. I think that it would be sufficient to say what the depth is even if others have done other measurements at this depth.

Answer: The first paper mentioned in this comment is important and now discussed in the revised paper. It demonstrates that by using adaptive optics particles can be trapped up to 166 μm in water. It is, however, our experience that trapping inside biological matter and retrieving quantitative information from such depths are more significantly challenging than trapping in water, in particular because of increased light scattering. The second paper

mentioned here by Favre-Bulle et al., Nature Com 2017 is a highly relevant and interesting paper. It was also discussed in the original manuscript and it describes trapping otoliths 150 μm in zebrafish. However, this paper does not probe for any quantitative material properties describing the material properties of biological tissue at this depth. We have revised our discussion to better incorporate this paper in the revised version of the manuscript.

Comment 1-6

Fig S3 is done under different conditions than the subsequent measurements in the Zebrafish

Answer: True, the experiments providing data for Figure S3 are carried out in a well-controlled in vitro setting, inside a Matrigel with well controlled concentrations, this serving as a model viscoelastic environment to which the *in vivo* results can be compared. The experimental conditions are as described in Ref 27, Borries et al. 2020. This has been specified in the caption of Figure S3 in the revised supporting information.

Reviewer #2

We thank Reviewer 2 for constructive comments which have served to improve the manuscript. We were happy that the reviewer considers the method as convincing and to have promise, and we particularly appreciated comment 2-4 as it led to novel and interesting results.

Comment 2-1

The abstract should include the quantified α values and a brief summary of the discussion/conclusion on the measured values.

Answer: This is a good idea, we have inserted a discussion of the obtained α values in the abstract.

Comment 2-2

The authors claim that the use of NIR laser allows the analysis of deep tissues. However, the location of the particles was assessed by confocal microscopy, which may negatively impact the advantage of the NIR laser. Please include a quantitative discussion on the achievable optical resolution with respect to the depth in the zebrafish body for visible and NIR wavelengths.

Answer: In the revised manuscript we have included a quantitative discussion on the optical resolution. We have also explained in more detail that the optical tweezers, based on an NIR laser, were used to monitor the thermal fluctuations of nanoparticles in living zebrafish. The tweezers were implemented in a Leica confocal microscope, however, during optical trapping the visible confocal lasers were turned off and the confocal was only employed to localize the sample and beads before trapping, using low laser powers and with low resolution. After the trapping experiments, zebrafish embryos were fixed and the yolk removed for a more careful examination of the sub-cellular location of the trapped nanoparticle in the foregut region using a different confocal microscope which was operated at high laser power, altogether providing significantly high resolution (~250 nm). In other words, the NIR laser was used for trapping (not for imaging) and the confocal microscope operating with high resolution was only used to image fixed specimen.

Comment 2-3

This means the measured viscoelasticity is always biased by how (where) nanoparticles stay in the tissue/cell. Is there a possible method to assess the contributions from actin filaments and other possible factors than microtubules or to find 'unbiased' viscoelasticity from your measurement?

Answer: The nanoparticles were micro-injected into the zebrafish at an early developmental stage, and the particles got distributed throughout the organism as the embryo developed. We analyzed and quantified the location of these particles within investigated cell populations (Figure S4 and Figure S5). Our analysis revealed that the vast

majority of particles investigated, regardless of cell type, are further away from the nucleus and from the periphery of the cell (actin cortex) than the amplitude of their thermal motion (~50 nm). Importantly, although it does not exclude all potential location bias, especially when we directly compare two or more different cells to each other, we have not observed location bias on the level of different cell populations. Hence, tracing their thermal motion did allow us to compare the viscoelastic properties of the cell types investigated.

For future experiments, one could consider disrupting actin via drug treatment. However, we decided not to do so because depolymerization of actin filaments via drug treatment compromises integrity of the polarized tissues (lateral plate mesoderm and developing gut tube) in the foregut region, which leads to drastic changes in cell morphology. For similar reasons, we have decided against using isolated cells, as despite having more control over the particle and its location, it would cause severe morphological and physiological changes.

Comment 2-4

The measurement was made at 28-30 hpf. How do the α values change at different (earlier) time points? When is the earliest time point to show the α value difference between the gut and the liver?

Answer: Prompted by this question we have performed additional measurements of α at 21-23 hpf in the endodermal region which will have developed by 28-30 hpf into liver and gut progenitors. Although it is possible to do measurements between 24-28hpf, the expression of biomarkers to identify liver and gut progenitors in the follow-up analysis is quite low. This prevented us from doing measurements at the 24-28hpf developmental stage, as we could not clearly assign the measured particle to a specific cell population. Therefore, we chose the earlier time-point, 21-23hpf, which is prior to overt lineage marker expression. The new results from 21-23 hpf are shown in Figure S8 and are discussed in the revised manuscript. The most important conclusion from these new experiments is that α from this earlier developmental stage of the endoderm falls between the two α -values characterizing, respectively, the liver and the gut progenitors 6 hours later. This new result is very interesting, as it suggests that the viscoelastic properties of both progenitor populations arise as the cells differentiate and that neither represents a default state. We thank the reviewer for providing this comment, leading us to this important addition to the manuscript.

Reviewer #3

We thank Reviewer 3 for constructive comments, which have helped us to improve the manuscript. We were delighted that Reviewer 3 found our results and the research question interesting. We particularly appreciate Reviewer 3's comments 3-2, 3-4, and 3-10, which have led to novel calculations of the complex shear moduli, using the suggested methodology and which, indeed, is based only on the frequency window in which thermal fluctuations can be assumed to dominate the signal.

Comment 3-1

Parameters (beta and kappa) from optical trap calibration in water seem to be uniformly applied to trapping experiments in-vivo, which will yield incorrect absolute values for any position or force calculations.

Answer: One advantage of the proposed method is that there is no need to calibrate the optical tweezers to obtain α . In order to determine α , we examine the scaling properties of the power spectrum, or of G' or G'' . For this, parameters like the spring constant of the optical trap (κ), or the conversion factor between voltage and displacement (β) are not required. We only need to know that voltage on the photodiode and tracer displacement are linearly correlated, which is indeed the case. All conclusions of the paper are based solely on α , therefore, it is an important aspect of the paper that a normal optical tweezers calibration procedure is not needed to find α . Reviewer 1 had essentially the same comment (Comment 1-4), hence, we had not explained this sufficiently well in the original manuscript. Hopefully this is clearer from the revised version.

No absolute position nor absolute force measurements are sought for nor are reported in the current study.

Comment 3-2

The assumption of thermal fluctuations within a small range (400-4000 Hz) seems to be applied uniformly over all frequencies, which will yield incorrect material properties.

Answer: The assumption that signals are dominated by thermal fluctuations, and not by non-equilibrium processes or photodiode filtering, is indeed valid only in the 400-4000 Hz interval. Therefore, we have changed the calculation of G' and G'' and we have adjusted Figure 2D, which now only shows this relevant interval. Our new calculation of the complex shear moduli is based on the method recently published by Nishi, K., Kilfoil, M. L., Schmidt, C. F., & MacKintosh, F. C. (2018) 'A symmetrical method to obtain shear moduli from microrheology', *Soft matter*, 14(19), 3716-3723. Notably, all results and conclusions remain the same after employing this method.

Comment 3-3

On page 4, the authors refer to the active-passive measurements — This is not just a different approach, but currently is one of the only approaches to get quantitative force

measurements in living cells using a QPD detection system. Further, there have been a large number of studies using similar active-passive approaches which results relevant to this study.

Answer: This study does not aim at providing absolute forces. We certainly agree with the reviewer that if absolute forces should be measured by QPDs, active-passive approaches would be preferable. This is discussed in the revised manuscript.

Comment 3-4

On page 5, the authors state they measure material properties using QPD detection of thermal fluctuations of optically trapped nanoparticles. The main issue in this study is the authors are measuring fluctuations due to all sources, not just thermal forces. Importantly, in a developing cell there are many active internally generated (non-thermal) forces that are contributing to the fluctuations measured. This is the main factor that makes their results of material properties difficult to interpret. For instance, the power-law scaling of the power spectrum cannot be attributed to solely the viscoelasticity of the cells but it is also affected by any activity in the cell. The power-law is thus a combination of both material properties and active fluctuations, not material properties alone.

- On page 6, they state the viscoelastic properties can be mapped by tracking thermal fluctuations. This is again the problem. The fluctuations are not solely thermal. Thus they cannot directly related the measured fluctuations to viscoelastic properties.

- Page 13, the authors summarize that their measurements show the viscoelastic properties can be quantified in vivo based on thermal fluctuations of nanoparticles. Again this is the problem - the fluctuations are not thermal overall. And while I agree the fluctuations above a certain frequency regime may be thermal, that frequency regime must first be verified and then this type of analysis (that relies on thermal fluctuations) cannot be applied outside that frequency range.

Answer: We had chosen the frequency range above 400 Hz, because the signal from non-thermal activity was reported to occur in living cells at frequencies below 300 Hz, whereas thermal fluctuations dominate the signal at higher frequencies. In the original manuscript we cited Ref. 22, Wessel et al., 2015, who used *Drosophila* embryos as a model system, for this statement. However, prompted by this comment, we have referenced now significantly more studies (see below) on this issue and expanded the discussion in the revised manuscript accordingly.

In biological systems, non-equilibrium processes occur in the low frequency region (<100 Hz) [2,3,4] and it has been shown that the fluctuation-dissipation theorem is satisfied for both active and passive measurements at frequencies higher than ~10 Hz [4,5,6,7]. Beside the internal activities within cells, other types of noise in low frequency regions (<300 Hz) could stem from mechanical drift in the experimental setup. Another issue to be aware of is the filtering effect from the photodiode. This can be characterized by a 3dB cut-off frequency, f_{3dB} , which has been measured for our system and reported to occur above 6

kHz [8,9]. For these reasons, we chose to fit the power spectra in a frequency interval between 400-4000 Hz to obtain α .

If absolute force measurements would have been the goal, it has proven possible to use oscillatory optical tweezers for this, even at frequencies where non-equilibrium processes contribute to the signal [10,11].

References:

- (1) Power spectrum of out-of-equilibrium forces in living cells: amplitude and frequency dependence, *Soft Matter*, 2009,5, 2947-2953
- (2) Probing force in living cells with optical tweezers: from single-molecule mechanics to cell mechanotransduction *Biophysical Reviews* volume 11, pages 765–782 (2019)
- (3) Active Mechanics Reveal Molecular-Scale Force Kinetics in Living Oocytes *Biophysical Journal*, Volume 114, Issue 7,2018,Pages 1667-1679
- (4) Equilibrium and out-of-equilibrium mechanics of living mammalian cytoplasm *Journal of the Mechanics and Physics of Solids*, Volume 107, 2017, Pages 284-293
- (5) Equilibrium physics breakdown reveals the active nature of red blood cell flickering *Nature Physics*, volume 12, pages 513–519 (2016)
- (6) Nonequilibrium Mechanics of Active Cytoskeletal Networks *Science* 315, 370–373 (2007)
- (7) Feedback-tracking microrheology in living cells *Science Advances*, 2017, Vol 3, Issue 9
- (8) Unintended filtering in a typical photodiode detection system for optical tweezers *Journal of Applied Physics* 93, 3167 (2003)
- (9) Power spectrum analysis for optical tweezers *Review of Scientific Instruments* 75, 594 (2004)
- (10) A comparative study of living cell micromechanical properties by oscillatory optical tweezers, *Opt. Express* 16, 8594-8603 (2008)
- (11) The Role of Vimentin Intermediate Filaments in Cortical and Cytoplasmic Mechanics, *Biophysical Journal*, Volume 105, Issue 7,2013, 1562-1568 (2013)

Comment 3-5

On page 11, the authors state they focus on the frequency regime where non-equilibrium processes are negligible between 400-4000Hz. It would be worth showing that this is the frequency range where non-eq processes are actually negligible. But assuming this is correct, all analysis of material properties must not be limited to this regime. Any frequencies outside this window cannot be analyzed in terms of the fluctuation dissipation theorem. Therefore, the G^ shown in Fig 2d is invalid outside of those frequency ranges.*

Answer: According to literature (see references above) it is fair to assume that non-equilibrium processes are negligible between 400-4000 Hz. We agree with Reviewer 3 that all analyses of material properties should be limited to this regime. For this reason, we have followed the advice of Reviewer 3 and have repeated all analyses of the complex shear moduli (see also answer to Comment 3-2). This new analysis has resulted in an updated Figure 2D in the manuscript, now showing only the relevant interval.

Comment 3-6

Further, force calibration must be done in-situ at the time of each measurement since the trap stiffness will depend on the local environment of each cell. Using a constant value for optical trap calibration confounds all results comparing between different cells and locations.

Answer: One advantage of the employed method is that there is no need to calibrate the optical tweezers to determine α . In the presented analyses we are not using any constant values for the parameters normally determined by calibration (e.g., kappa or beta) – these parameters are not required to retrieve α . See also the answer to comment 3-1.

Comment 3-7

Page 15, fig 3, shows the non-dimensional alpha parameter. This value is the only quantitative metric I see in the study that allows comparison between different foregut regions because the absolute values of the spectrum do not matter (which are affected by the calibration). The non-dimensional scaling should still be valid. If the goal is to compare material properties, this value can at least accomplish a relative comparison within the region of 400-4000Hz.

Answer: Yes, this is true. We agree with the description provided by Reviewer 3 and have changed the wording throughout the manuscript to clarify that the goal of this study is to perform a relative comparison within the region of 400-4000Hz.

Comment 3-8

Page 19, the authors state that this is a significant improvement over particle tracking in video recording, and that is certainly true. But it might be a good idea to cite others who have done laser tracking with a QPD in cells previously such as: Yamada, S., Wirtz, D., & Kuo, S. C. (2000). Mechanics of living cells measured by laser tracking microrheology. Biophysical journal, 78(4), 1736-1747. And many others.

Answer: Yes, a number of groups have performed laser tracking of tracers in single cells. The major difference to the current study is that we are demonstrating this for cells deep within a living organism. We have included and discussed the suggested study by Yamada et al., who were among the first scientists to achieve this and who are credited in the revised manuscript.

Comment 3-9

SM page 1, it looks as if the authors calibrate their value of beta (and subsequently kappa) using a bead trapped in water. This is clearly not applicable in their living samples. The values of beta and kappa will not only differ from in water greatly, they will depend on space and time inside of the living organism. If this is how the calibration was done, then it seems nearly impossible to interpret the absolute values of any of the measurements. If this is not how the calibration was done, the paper needs to be revised extensively to describe how force calibration was carried out.

Answer: As discussed above (answers to comments 3-1 and 3-6) one advantage of the proposed method is that there is no need to calibrate the optical tweezers to obtain α . All conclusions on the biological system are based on α .

The reason we presented a calibration in water in the previous version of the manuscript was to estimate the absolute values of G' and G'' in Figure 2D. For this, we used the calibration parameters obtained from trapping a similar particle in water. In the revised manuscript we have decided not to present an estimate of the absolute value of G' and G'' , because it is not required, as we are only interested in the scaling properties of G' and G'' .

In the revised manuscript we provide G' and G'' in arbitrary units in Figure 2D and we removed the calibration in water.

Comment 3-10

*SM page 2, equation 4 is only valid in thermal equilibrium since it is based on the fluctuation dissipation theorem. Assuming fluctuations are thermal from 400-4000 Hz, the authors can only use this equation in that range to calculate the imaginary part of the response function. But the authors use this equation to cover 10Hz - 10kHz, which is well outside the range of thermal fluctuations. Since 400-4000Hz limits them to one decade in frequency, it is also questionable if the scaling can be even called a power-law since an exponential decay over 1 decade looks very similar. Further, the G' values will be over an even smaller frequency range (400 - 1000 Hz) due to the Kramers-Kronig relation. If the authors wish they can regain some of their range in G' by using more recent methods: Nishi, K., Kilfoil, M. L., Schmidt, C. F., & MacKintosh, F. C. (2018). A symmetrical method to obtain shear moduli from microrheology. *Soft matter*, 14(19), 3716-3723.*

Answer: We thank the reviewer for this comment and suggestion. We were not aware of this work prior to Reviewer 3's comments. In the revised version of the manuscript, we

therefore repeated our entire data analyses of the complex shear modulus using the suggested method: A symmetrical method to obtain shear moduli from microrheology. *Soft matter*, 14(19), 3716-3723. As indicated, the method is used only in the relevant frequency range. The entire section in the supporting information describing the calculation of the complex shear modulus has been updated accordingly.

This new analysis changed the values only slightly and, importantly, did not change any conclusions regarding the biological system.

Comment 3-11

The paper is missing reference to the wide body of literature on non-equilibrium fluctuations in cells, which are precisely the studies that show the authors approach does not yield true material properties. I encourage the authors to include some of these citations from the last 5 years which showed an explosion of studies in this area.

Answer: More references to literature on non-equilibrium systems have been discussed, kindly see answer to comment 3-4 above and the references given there.

Comment 3-12

Page 8, Figure 2 is pixelated and difficult to read. Why is G'' omitted for water? Panels B,C, do not show an observable plateau in the low frequency range for particles in the cells — This suggests either that the particles are not trapped or they exhibit active fluctuations in the low-frequency.

Answer: Thank you for pointing this out, Figure 2 in the revised version of the manuscript has higher quality (however, there are limits to how large figures we can upload at the revision stage). G' and G'' for water are provided in Supporting Figure S2, which only serves the purpose of introducing the method. At the relatively low laser powers used here during optical trapping inside a living cell there is usually not an observable plateau region within the depicted frequency window, for examples see, e.g., Ref 8 Tolic-Nørrelykke et al., PRL 2004 or Selhuber-Unkel et al., *Physical Biol.* Vol 6 p 025015, 2009. See also answers to Comments 3-1 and 3-6, knowledge of kappa is not needed for the analysis.

Comment 3-13

Page 11, they estimate the heating affects based on other studies and state they are negligible. However this is something that can be verified very easily in their own system. The authors should do so, and likely already have the data (repeated measurements in same spot) to check this.

Answer: To address the possible heating effects, which would be highly important for living organisms, we included Ref 49, Peterman et al., *Biophys J* 2003. In addition, our own group has conducted a significant amount of work on the heating of nanoparticles (see for instance Bendix et al. *ACS Nano* vol 4 p 2256 2010 or Jauffred et al, *Chemical Reviews* vol 119 p 8087 2019). This body of work has given us confidence that the temperature

increases in the current experiment are as stated in the manuscript. In addition, we have followed the suggestion of Reviewer 3 and conducted repeated measurements at the same location in a living zebrafish. Data from repeated measurements are shown below in Figure 1. The conclusion is that repeated measurements do provide the same result.

Figure 1: Repeated probing, in form of three measurements conducted with 3 seconds intervals, in the same location in a living zebrafish generate similar power spectra with identical scaling exponents.

REVIEWERS' COMMENTS:

Reviewer #1 (Remarks to the Author):

The revised manuscript has adequately addressed all the comments of the three referees. I am satisfied with the implemented changes and believe that the paper can be published without further changes. However I have one comment to the last comment made by the authors to the Reviewer #3 comments. The authors state: "If absolute force measurements would have been the goal, it has proven possible to use oscillatory optical tweezers for this, even at frequencies where non-equilibrium processes contribute to the signal [10,11]." In my view this is a bit of a questionable statement as the use of the oscillatory tweezers inside the cells can be made difficult as it has to be done in rather small/tight spaces which means that the wall effects will be very significant and make the measurement rather difficult. I would suggest that the authors add something like that to their statement. Also they can suggest to use for the absolute force measurements rotational optical tweezers.

Reviewer #3 (Remarks to the Author):

The authors have satisfactorily addressed all my comments in their response and revised manuscript.